# On the Efficiency of Diffusion Models in Generating Plausible Designs

## Abstract

Diffusion-based generative models have huge potential in creating novel structural images in generative design where the user heavily values the design plausibility, e.g, no floating material or missing part. However, such models often require many denoising steps to achieve satisfactory plausibility, resulting in high computation costs; when using much fewer steps, we can not ensure plausibility. This paper addresses this trade-off and proposes an efficient training and inference method that can achieve the same or better plausibility than existing models while reducing the sampling time. We determine the noise schedule based on the evolution of pixel-value distributions in the forward diffusion process. Compared to previous models, e.g., DDPM and EDM, our method concentrates the noise schedule at a range of noise levels that highly influence the structural modeling and hereby achieves high efficiency in inference without compromising the visual quality or design plausibility. We apply this noise schedule to the EDM method on two structural data sets, BIKED and Seeing3DChairs. On BIKED images, for instance, our noise schedule significantly improves the quality of generated designs: the rate of plausible designs from 83.4% to 93.5%; Fréchet Inception Distance from 7.84 to 4.87, compared to EDM.

## 1 Introduction

Deep Generative Models (DGMs) have emerged as a powerful algorithm for enabling Generative Design, hereby modeling and exploring design spaces. This approach has also been employed for structural designs and shown its potential in synthesizing complex structures (Regenwetter et al., 2022; Chen & Ahmed, 2020; Chen et al., 2020; Fan et al., 2023). In design generation tasks, especially for structures, DGMs may generate implausible designs, e.g., bicycles with an extra handle, missing wheel or an invalid layout as shown in the first-row of Figure 1, which need to be avoided. Hence, for design generation tasks, we also focus on the plausibility of generated images, in addition to the commonly used metrics in image generation, e.g., Fréchet Inception Distance (FID) (Heusel et al., 2018).

Generative adversarial networks (GANs) (Goodfellow et al., 2014; Radford et al., 2016; Karras et al., 2019) and diffusion models (Ho et al., 2020; Song et al., 2020; Dhariwal & Nichol, 2021; Karras et al., 2022) have been widely applied in the image generation task. Recently, diffusion-based generative models have been reported to surpass GANs in various image synthesis tasks (Ho et al., 2020; Dhariwal & Nichol, 2021; Karras et al., 2022). Despite the success of diffusion-based models, several issues exist to address when generating design structures. For instance, Denoising Diffusion Probabilistic Models (DDPM) (Ho et al., 2020) can generate structure images with high visual quality; however, it suffers from a slow generation speed due to the usage of an excessive number of denoising steps (Kong & Ping, 2021; Song et al., 2020; Karras et al., 2022). As a remedy, the Denoising Diffusion Implicit Model (DDIM) (Song et al., 2020) greatly reduces the denoising steps, which, however, compromises the visual quality. More recently, the well-known EDM method (Karras et al., 2022) introduces a probability distribution to sample the noise levels in the denoising process, which can improve the generation speed and maintain a satisfactory visual quality, especially for rendering image details, e.g., human hair curls and skin pores. However, when assessing the plausibility of the generated design images, we observe that the EDM often produces structurally implausible images.

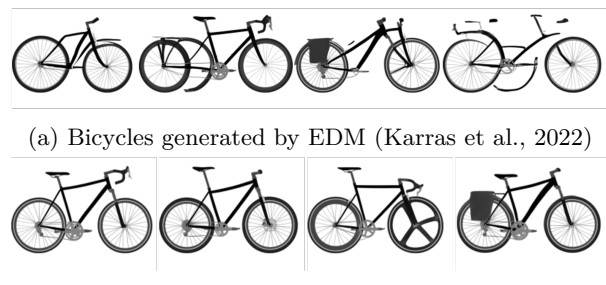

(a) Bicycles generated by EDM (Karras et al., 2022)

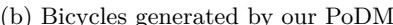

(b) Bicycles generated by our PoDM

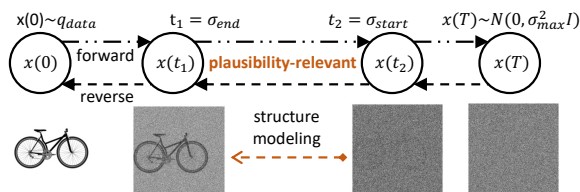

(c) Forward and reverse processes of diffusion models in directed graphs, with a highlight on the plausibility-relevant range of noise levels, i.e., $[\sigma_{\text{end}}, \sigma_{\text{start}}]$

Figure 1: Generated bicycle designs. Our work aims to minimize the proportion of implausible designs generated by focusing on a certain range of noise levels.

For the denoising process, recent research has explored the impact of the noise schedule on the properties of generated images (Nichol & Dhariwal, 2021; Song & Ermon, 2020; Song et al., 2021; Karras et al., 2022). For example, while lower noise levels can enhance the quality of image details (Karras et al., 2022), higher noise levels can affect the diversity of generated results (Song & Ermon, 2020). In this paper, we discover that in using diffusion models, there exists a plausibility-relevant range of noise levels that predominantly affect the plausibility of the images (see Figure 3). More importantly, this range can be determined by the evolution of pixel-value distributions in the forward diffusion process (3.2). We demonstrate the importance of the plausibility-relevant range in Figure 1c. To determine it, we simulate the forward diffusion process on real structural designs and trace the distribution of pixel values as the noise level increases. We observe that the disappearance of the structural signal has a clear corresponding phase in the development of pixel-value distributions.

Taking this observation, we modify the training and generation procedures of EDM to prioritize sampling the noise levels in the plausibility-relevant range (see Figure 2a for an illustration), resulting in a new method, Plausibility-oriented Diffusion Model (PoDM). We experimentally test PoDM on two datasets, the BIKED dataset (Regenwetter et al., 2021) and the Seeing3DChairs dataset (Aubry et al., 2014), in terms of the following metrics: (1) Fréchet Inception Distance (FID) (Heusel et al., 2018) for visual quality; (2) Plausible Design Rate (PDR), the proportion of plausible designs for 1 000 evaluated images (see Section 4.2 for details); and (3) Frames per Second (FPS) for generation speed. On the BIKED data, our PoDM outperforms EDM on PDR: 93.5% (PoDM) vs. 83.4% (EDM) and on FID: 4.87 (PoDM) vs. 7.84 (EDM), while achieves comparable FPS with EDM. Compared to DDPM, PoDM has a comparable PDR value (PDR: 94%, FID: 11.77) but is ca. 15× faster in terms of FPS. We observe a similar comparison in the Seeing3Dchairs data.

Lastly, we further test the performance of PoDM in incorporating modern image-editing methods, e.g., inpainting, interpolation via latent space and point-based dragging, and hereby manipulating structure.

## 2   Related work

**DGM-driven structure generation** The field of Deep Generative Models (DGMs) is developing at an astonishing speed, as some models can generate novel data that is able to fool humans, especially in image synthesis (Karras et al., 2019; 2020; Wu et al., 2019; Karras et al., 2022). Novel research proposes to employ DGMs' generation power to generate structure designs and yields models with decent performance, e.g., PaDGAN (Chen & Ahmed, 2020) and BézierGAN (Chen et al., 2020) for the UIUC Airfoil shapes (University of Illinois at Urbana-Champaign, 2022); and Self-Attention Adversarial Latent Autoencoder (SA-ALAE) (Fan et al., 2023) for complex designs of automotive parts. Meanwhile, Regenwetter et al. (Regenwetter et al., 2021) introduces the BIKED dataset to challenge DGMs in generating complex structural designs, which is a suitable dataset for exploring the performance of DGMs in structure generation. However, to the best of our knowledge, there have not been any convincing results in generating BIKED images. The only known result to us is achieved by (Regenwetter et al., 2021) with Variational Autoencoders (VAEs) (Kingma & Welling, 2013), but the generated designs are barely recognizable.

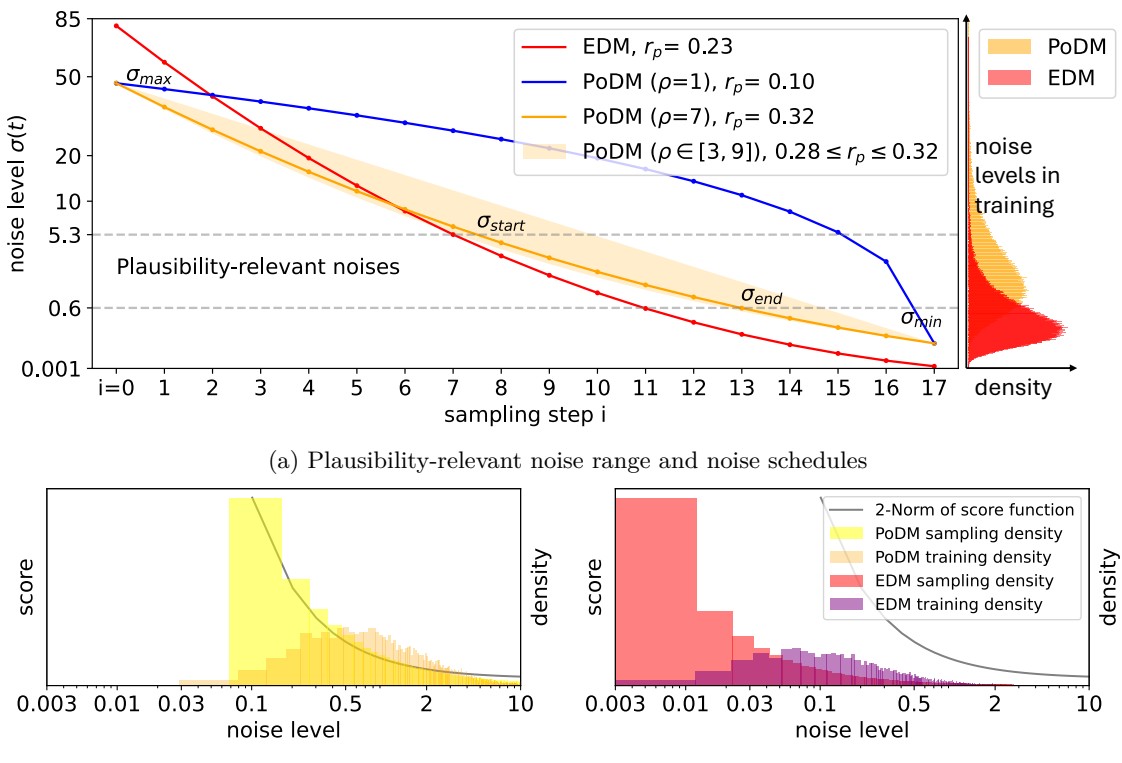

(a) Plausibility-relevant noise range and noise schedules

(b) Magnitude of the score function and density of noise schedules

Figure 2: (a) We showcase the plausibility-relevant noise range $[\sigma_{end}, \sigma_{start}]$ (*dashed interval*) computed on the BIKED dataset with the techniques proposed in Section 3.2. For the denoising process, our PoDM method takes an exponentially decaying schedule (*blue and orange curves*) with a parameter $\rho$ controlling the decay rate. PoDM method determines the minimal/maximal noise levels of the schedule ($\sigma_{min}$ and $\sigma_{max}$, respectively) based on $[\sigma_{end}, \sigma_{start}]$ (Equation 8). As for the noise levels sampled in the training process, our PoDM method uses a log-normal density concentrating on $[\sigma_{end}, \sigma_{start}]$ (*the orange histogram shown vertically*), in contrast to the fixed distribution used in EDM (*the red histogram*). (b) For PoDM and EDM, we depict the magnitude of the score function (Equation 2) at different noise levels, which is compared to the density of the noise levels in the training and denoising/sampling steps. The noise density in the sampling process of PoDM matches closely with the score function.

**Diffusion-based generative modeling** Diffusion models (DM) (Sohl-Dickstein et al., 2015) present a novel idea for capturing data distribution and image generation. DMs did not attract much attention until the convincing implementation of the Denoising Diffusion Probabilistic Model (DDPM) (Ho et al., 2020), which leverages tremendous sampling time to generate images with quality comparable to GANs. A further developed diffusion model, Denoising Diffusion Implicit Model (DDIM) (Song et al., 2020), improves generation speed by trading off image quality. Meanwhile, in the domain of score-based generation, Song et al. (Song et al., 2021) proposes to use a Stochastic Differential Equation (SDE) for the forward process and a corresponding reverse-time SDE for sampling, which allows continuous diffusion processes. SDE derives a deterministic sampling process based on a corresponding ordinary differential equation (ODE), that enables identifiable encoding-decoding and more importantly flexible data manipulation via latent space. Then, Karras et al. (Karras et al., 2022) clean the design space of diffusion-based generative models and propose a novel framework, denoted as EDM. EDM achieves a new state-of-the-art performance on the generation of CIFAR-10 (Krizhevsky et al., 2010) and ImageNet-64 (Deng et al., 2009).

**Diffusion-based image editing** Diffusion-based generative models have been introduced in controlling generation and data manipulation, e.g., interpolation via latent space (Ho et al., 2020; Song et al., 2021; 2020; Dhariwal & Nichol, 2021), free-form inpainting (Song et al., 2021; Lugmayr et al., 2022) and point-

based dragging (Shi et al., 2023). These image editing methods tend to be applied on natural images, e.g., CelebA (Liu et al., 2015), LSUN bedroom images (Yu et al., 2015) and ImageNet (Deng et al., 2009).

# 3 Plausibility-oriented Diffusion Modeling

In perturbation experiments with increasing noise levels on BIKED (Regenwetter et al., 2021), as the results shown in 3.2, we observe that the structural design fades away over a range of noise levels, and this range is clearly identifiable in the development of the pixel-value distribution. Same observation has been noticed also on FFHQ Karras et al. (2019), Seeing3DChairs (Aubry et al., 2014) and Shoes (Yu et al., 2016). We think that this range is the plausibility-relevant range of noise levels. Here, we propose two techniques to help with determining this range statistically. In 3.3, based on the EDM method (Karras et al., 2022) we present novel training and sampling procedures that target their efforts to the determined range of noise levels.

## 3.1 Background

We built up our contribution based on the stochastic differential equation (SDE) model of the diffusion process (Song et al., 2021; Karras et al., 2022). Given a data point $\boldsymbol{x} \in \mathbb{R}^d$, we corrupt it with the following forward Itô SDE (Oksendal, 2013):

$$d\boldsymbol{x} = f(\boldsymbol{x}, t)dt + g(t)dB_t, \tag{1}$$

where $f \colon \mathbb{R}^d \times [0, T] \to \mathbb{R}^d$ is the drift vector, $g \colon [0, T] \to \mathbb{R}$ is the dispersion coefficient, and $B_t \in \mathbb{R}^d$ is the standard Brownian motion. Notably, $f$ and $g$, are pre-determined by the user and have no trainable parameters. The corresponding reverse-time/backward SDE is (Anderson, 1982):

$$d\boldsymbol{x} = [f(\boldsymbol{x}, \tau) - g(\tau)^2 \nabla_{\boldsymbol{x}} \log p(\boldsymbol{x}, \tau)]d\tau + g(\tau)d\mathbf{B}_{\tau}, \tag{2}$$

where $\tau$ goes from $T$ to $0$, $p(\boldsymbol{x}, \tau)$ is the probability density of $\boldsymbol{x}$ at $\tau$ in the forward process, and $\nabla_{\boldsymbol{x}} \log p(\boldsymbol{x}, \tau)$ is known as the score function. The flow of probability mass in Equation 2 can be equivalently described by an ordinary differential equation (ODE) (Maoutsa et al., 2020; Song et al., 2021; Karras et al., 2022):

$$\frac{d\boldsymbol{x}}{d\tau} = f(\boldsymbol{x}, \tau) - g(\tau)^2 \nabla_{\boldsymbol{x}} \log p(\boldsymbol{x}, \tau). \tag{3}$$

We follow the choice of the drift and dispersion terms in (Karras et al., 2022) (a.k.a. EDM):

$$f(\boldsymbol{x}, \tau) = 0, \quad g(\tau) = \sqrt{2\sigma(\tau)}, \quad \sigma(\tau) = \tau,$$

and the score function is approximated by $\nabla_{\boldsymbol{x}} \log p(\boldsymbol{x}, \tau) = (D_\theta(\boldsymbol{x}; \sigma(\tau)) - \boldsymbol{x})/\sigma(\tau)^2$, where $D_\theta$ is a neural network trained on samples drawn from the forward SDE (see (Karras et al., 2022) for details on the loss function). Due to the above choice, $\sigma$ and $\tau$ are interchangeable henceforth. To solve/sample from Equation 3, an $N$ time-step discretization is used with the following noise schedule: $\sigma_N = 0, \forall i \in [0..N-1]$:

$$\sigma_i = \left(\sigma_{\max}^{\frac{1}{\rho}} + \frac{i}{N-1}(\sigma_{\min}^{\frac{1}{\rho}} - \sigma_{\max}^{\frac{1}{\rho}})\right)^{\rho}, \tag{4}$$

where $\sigma_0 = \sigma_{\max}$ and $\sigma_{N-1} = \sigma_{\min}$. EMD recommends the setting: $\sigma_{\min} = 0.002, \sigma_{\max} = 80, \rho = 7$. In the stochastic sampling procedure, we denote by $\boldsymbol{x}_i$ the data point obtained at $\sigma_i$. We first increase the noise level slightly and perturb $\boldsymbol{x}_i$:

$$\boldsymbol{x}_i' = \boldsymbol{x}_i + \sqrt{\hat{\sigma}_i^2 - \sigma_i^2} \mathcal{N}(\mathbf{0}, S_{\mathrm{noise}}^2 \mathbf{I}) \tag{5}$$

$$\hat{\sigma}_i = \sigma_i \left(1 + \mathbb{1}_{[S_{\min}, S_{\max}]}(\sigma_i) \min\left(S_{\mathrm{churn}}/N, \sqrt{2} - 1\right)\right), \tag{6}$$

where $S_{\mathrm{churn}}$ controls the degree of randomness in sampling: $S_{\mathrm{churn}} = 0$ realizes deterministic generation. Afterwards, we apply the reverse-time ODE ( Equation 3) with $\boldsymbol{x}_i'$ from $\hat{\sigma}_i$ to $\sigma_{i+1}$. The default settings of stochastic sampling are: $S_{\mathrm{churn}} = 40, S_{\min} = 0.05, S_{\min} = 50, S_{\mathrm{noise}} = 1.003$. The training data of $D_\theta(\boldsymbol{x}; \sigma(t))$ are sampled from  Equation 1 with a log-normal distribution: $\ln(\sigma) \sim \mathcal{N}(P_{\mathrm{mean}}, P_{\mathrm{std}}^2)$. In (Karras et al., 2022), the following empirical setting is suggested: $P_{\mathrm{mean}} = -1.2, P_{\mathrm{std}} = 1.2$.

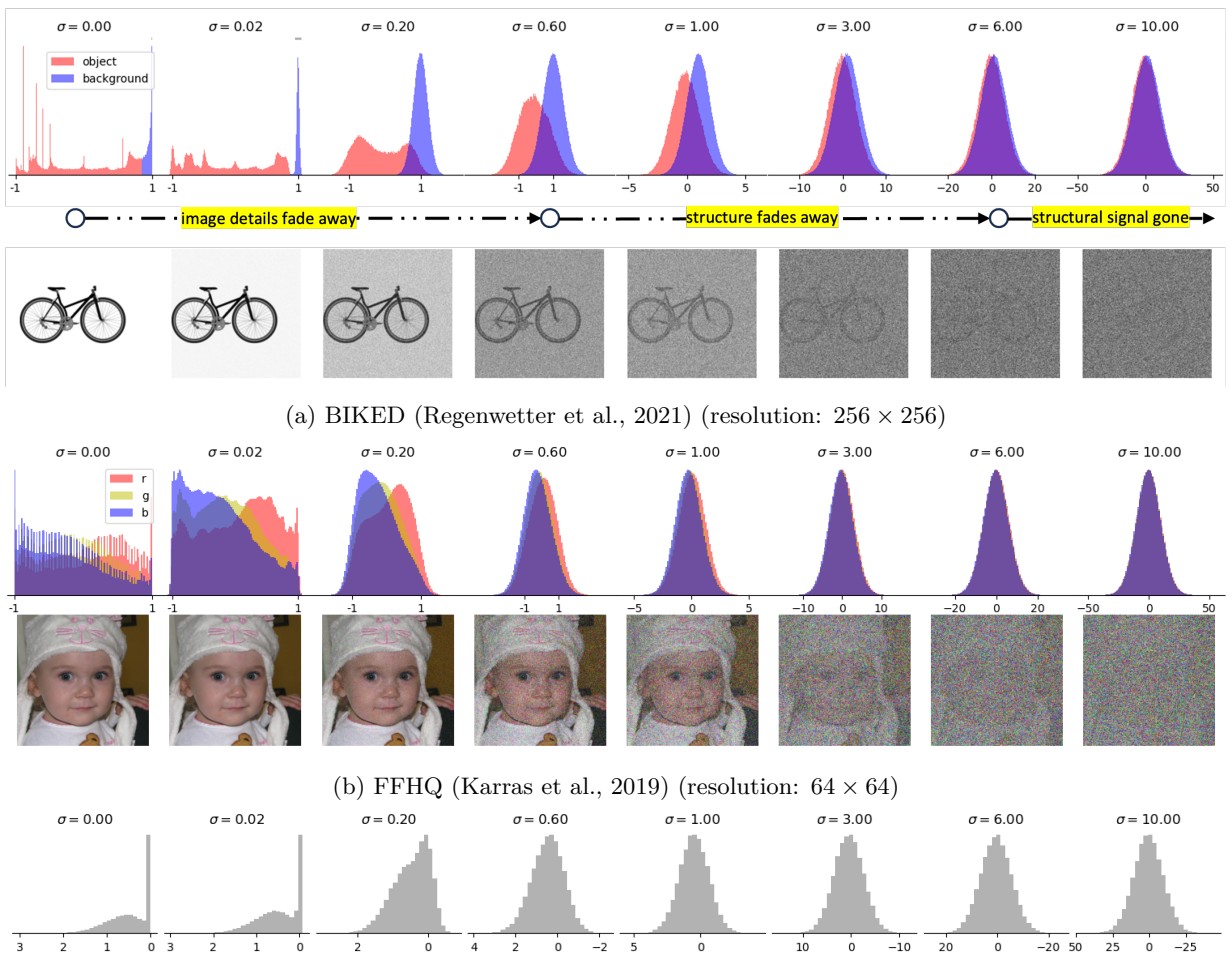

(a) BIKED (Regenwetter et al., 2021) (resolution: $256 \times 256$)

(b) FFHQ (Karras et al., 2019) (resolution: $64 \times 64$)

(c) Distribution of latent features extracted from the BIKED data (Regenwetter et al., 2021) with a convolutional autoencoder.

Figure 3: Evolution of the pixel-value distribution in perturbation experiments for (a) the BIKED data, a set of design structures, and (b) FFHQ, a real-world data set. In (c), we show the evolution of the latent-value distribution after encoding the BIKED data into a 64-dimensional space with a convolutional autoencoder.

Table 1: Grid search results of the parameter $\rho$ of the noise schedule on the BIKED data w.r.t. three performance metrics. The *column* $r_p$ measures the proportion of noise levels falling into the plausibility-relevant range, which is strongly correlated with the performance metrics.

| $\rho$ | $r_p$ | FID↓ | DPS↑ | PDR↑ |
|---|---|---|---|---|
| 1 | 0.10 | 12.67 | 4.70 | 82.8% |
| 3 | 0.28 | 5.64 | 4.81 | 88.6% |
| 5 | 0.31 | 5.25 | 4.88 | 89.6% |
| 7 | 0.32 | **4.87** | **4.90** | **93.5%** |
| 9 | 0.32 | 5.18 | 4.87 | 91.5% |

## 3.2 Noise range relevant to plausibility

We conjecture that in the forward/backward diffusion process, there exist a range of noises that plays a crucial role in object construction. We validate it by running the forward process with fine-grained noise

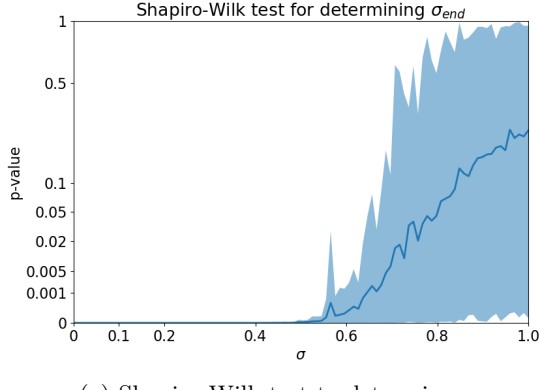

(a) Shapiro-Wilk test to determine $\sigma_{\text{end}}$

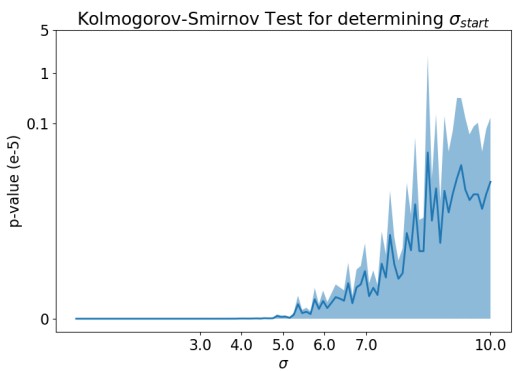

(b) Kolmogorov-Smirnov test to determine $\sigma_{\text{start}}$

Figure 4: On the BIKED data, (a) we illustrate **Technique 1** by plotting the $p$-value of the Shapiro-Wilk test computed from 100 randomly picked images, which are perturbed by different noise levels; (b) for **Technique 2**, we show the Kolmogorov-Smirnov test between the bicycle's and background's pixel-value distribution as a function of the noise levels. In both plots, the variability is depicted as the *shaded region*.

levels: starting with a source image $\boldsymbol{x}(0)$ (the pixel values are standardized to $[-1, 1]$ before adding the Gaussian noise), we keep adding small Gaussian noises thereto until its pixel-value distribution becomes indistinguishable from a Gaussian: $\boldsymbol{x}(t) = \boldsymbol{x}(0) + 0.1t \times \mathcal{N}(\mathbf{0}, \mathbf{I})$. In Figure 3a, we show the pixel-value distribution at intermediate time steps computed from 100 images sampled from BIKED (Regenwetter et al., 2021). The BIKED images consist of a single bicycle object and a monotone background. We depict the pixel-value distribution separately for the object and the background (the red and blue histograms, respectively). We observe three phases: (1) from the beginning to the first time when the pixel-value distribution of the bicycle converges to a Gaussian, i.e., $\sigma \in [0, 0.6]$. The pixel-value distribution of the bicycle is substantially different from that of the background in this phase; (2) the bicycle structure starts to fade away while the pixel-value distribution thereof overlaps more with the background, i.e., for $\sigma \in [0.6, 6.0]$; (3) the bicycle structure almost disappears for $\sigma > 6$. Empirically, we assume that the noise range ($\sigma \in [0.6, 6]$ in Figure 3) in which the bicycle structure fades away determines the plausibility of the generated structural. We denote this noise range as $[\sigma_{\text{end}}, \sigma_{\text{start}}]$ and propose two techniques to determine this interval.

**Technique 1** *Choose $\sigma_{end}$ to be the noise level when the curve of p-value in Shapiro-Wilk test begins to diverge.*

In a sampling process, $\sigma_{\text{end}}$ is the noise level at which the generation is structurally finalized and object pixel values remain normally distributed. Denoising with noise levels of $\sigma \leqslant \sigma_{\text{end}}$ performs a refinement, during which the backward process approximates the object pixel values from a normal distribution to a local data distribution. To estimate $\sigma_{\text{end}}$, we propose to use the Shapiro-Wilk test (SHAPIRO & WILK, 1965) to track the distribution of the object's pixel values during the perturbation test. As displayed in Figure 4a, the measured $p$-value increases with the noise level. In practice, specific dataset might exist, where the object pixel-value distribution is already Gaussian distributed, here we set a minimum limitation of $\sigma_{\text{end}}$ to be 0.08 (converting the default parameter values of EDM to our parameter system, we obtain the value $\sigma_{textend} = 0.08$).

**Technique 2:** *Choose $\sigma_{start}$ to be the noise level when the curve of p-value in Kolmogorov-Smirnov test begins to diverge.*

In the synthesis of structural design images, $\sigma_{\text{start}}$ is the noise level at which the structural formation begins, i.e., pixels of objects begin to distinguish themselves from pixels of the background. To measure such a difference, we first approximate the pixel-value distributions of the object and background with Gaussians, respectively, and then conduct the Kolmogorov-Smirnov test between the two Gaussian approximations. As shown in Figure 4b, $\sigma_{\text{start}}$ is taken when the curve of $p$-value measured in the Kolmogorov-Smirnov test begins to diverge.

Our work gives an insight into defining the plausibility-relevant range of noise levels so that the training and sampling effort can prioritize this range. By implementing our two techniques, for BIKED images, we determine $\sigma_{\text{end}}$ to be 0.6 and $\sigma_{\text{start}}$ to be 5.3. This observation can also be seen in real-world images, such as those from FFHQ (Karras et al., 2019). Since it is difficult to automatically separate faces from backgrounds, we first track the distribution of pixel values in each RGB channel without distinguish pixels of faces and of backgrounds: as seen in Figure 3b, the distribution of pixel values in each color channel starts from an irregular distribution, gradually converges to a Gaussian distribution (at about $\sigma = 0.60$), and then overlaps each other at a certain noise scale (at about $\sigma = 0.60$). To enhance the observation and the applicability of our theory, in Appendix A.1, we conduct this experiment on more datasets, e.g., Seeing3DChairs (Aubry et al., 2014) and Shoes (Yu et al., 2016), and in Appendix A.2, we observe the same phases in each color channel when we manually separate the FFHQ faces from their backgrounds. To showcase that the above observation can also be seen in the latent diffusion models (Rombach et al., 2022), we first encode 100 BIKED images with a convolutional autoencoder trained on BIKED into a 64-dimensional latent space and then show the evolution of the latent-value distribution (see Figure 3). We observe a pretty similar trend in the latent-value distribution as with the pixel-value distribution. Note that the plausibility of design images, i.e., BIKED Regenwetter et al. (2021) and Seeing3DChairs (Aubry et al., 2014), is easier to assess, the background can be automatically separated from the main object, and therefore the plausibility-relevant noise range can be more accurately estimated. Thus, this work will focus on structural designs to demonstrate the efficiency of our proposal.

### 3.3 Plausibility-oriented training and generation procedures

We modify the training and generation procedures so that our diffusion model can concentrate on the plausibility-relevant range of noise levels.

**Noise density in training** For the structure images, the generation of the structure takes place mostly in the noise range $[\sigma_{\text{end}}, \sigma_{\text{start}}]$ while the noise levels that are too small or large have marginal effects on the plausibility of the final outcome. Hence, it is sensible to sample more noise levels in this interval from the forward SDE. We propose the following log-normal distribution to sample noise levels:

$$\ln(\sigma) \sim \mathcal{N}(\mu, \, \zeta^2), \; \mu = \frac{\ln(\sigma_{\text{start}}) + \ln(\sigma_{\text{end}})}{2}, \; \zeta = \frac{\ln(\sigma_{\text{start}}) - \ln(\sigma_{\text{end}})}{2}, \tag{7}$$

which implies $\Pr(\sigma \in [\sigma_{\text{end}}, \sigma_{\text{start}}]) \approx 68\%$. In this method, the majority of the noise levels are drawn in $[\sigma_{\text{end}}, \sigma_{\text{start}}]$ while we have ca. 32% probability to sample noise levels at the beginning and the end of the forward process.

**Noise schedule for image generation** In the backward diffusion process (Equation 3), there are two important factors w.r.t. the noise levels: (1) the noise range $[\sigma_{\min}, \sigma_{\max}]$ in which we apply the ODE (Equation 3) and (2) the decaying noise schedule. For the former, we determine the range based on the training noise density as follows: Equation 7 implies that the score function $\nabla_{\boldsymbol{x}} \log p(\boldsymbol{x}, \sigma)$ is trained on noise levels drawn almost in $[\mu - 3\zeta, \mu + 3\zeta]$, i.e., $\Pr(\log(\sigma) \in [\mu - 3\zeta, \mu + 3\zeta]) \approx 99.7\%$. Therefore, when sampling new images, applying the reverse-time ODE out of $[\mu - 3\zeta, \mu + 3\zeta]$ requires the score function to extrapolate, which we have no guarantee about its accuracy. Hence, we set

$$\log \sigma_{\min} = \mu - 3\zeta = 2 \log \sigma_{\text{end}} - \log \sigma_{\text{start}} \;, \tag{8}$$

$$\log \sigma_{\max} = \mu + 3\zeta = 2 \log \sigma_{\text{start}} - \log \sigma_{\text{end}} \;. \tag{9}$$

For the latter, we follow the exponential decay in Equation 4, where, in addition, we tune the hyperparameter $\rho$ for the BIKED dataset. In Table 1, we summarize the tuning results from a simple grid search, where $\rho = 7$ is the best setting. Also, we observe that the performance metrics (e.g., FID, DPS, and PDR) are quite sensitive to $\rho$, suggesting that tuning this parameter is necessary across different structural image data. Moreover, we calculate the proportion of the noise levels $\{\sigma_{N-1}, \ldots, \sigma_0\}$ (determined by Equation 4) falling into the plausibility-relevant range $[\sigma_{\text{end}}, \sigma_{\text{start}}]$, which we call the prioritization density $r_p$. It measures how much training effort is targeted at the structure modeling. In Figure 2 and Table 1, we show $r_p$ with varying hyperparameter $\rho$, and we observe that the performance metrics are positively related to it.

We demonstrate a theoretical insight into the noise schedule in Figure 2b. On the BIKED data, we depict the norm of score function $\nabla_{\vec{x}} \log p(\vec{x}, \sigma(t))$ over noise levels. Comparing it with the histograms of the noise schedule, we see that PoDM's noise scheduling in sampling/denoising is concordant with the score function, meaning that finer steps of simulating the backward ODE/SDE (see Equation 3) are taken where the norm of the score function is large. We argue that it is sensible to do so since the score function is the major drift term of the backward ODE/SDE.

## 4 Evaluation and results

In this section, we compare our Plausibility-oriented Diffusion Model (PoDM) with other cutting-edge diffusion-based models, i.e., vanilla DDPM (Ho et al., 2020), DDIM (Song et al., 2020) and EDM (Karras et al., 2022). We also include SA-ALAE (Fan et al., 2023) in the comparison, considering that SA-ALAE leverages adversarial training and was recently proposed to address the generation of complex structural designs.

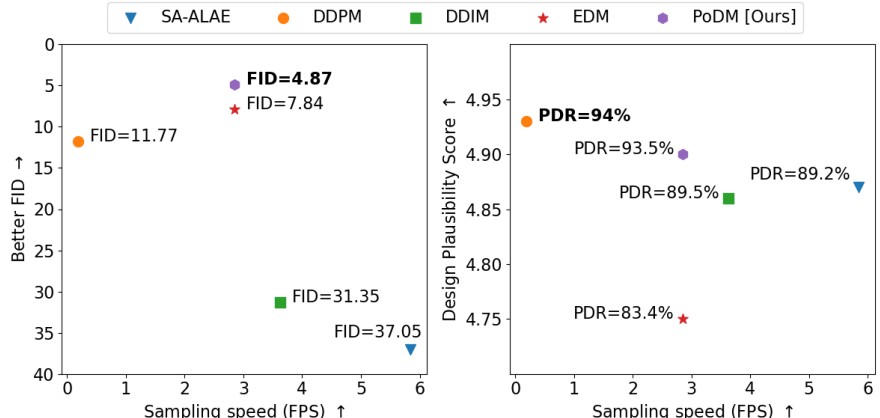

(a) On the BIKED data, we show two performance views of the five considered generative models: FID vs. FPS and design plausibility score (DPS) vs. FPS.

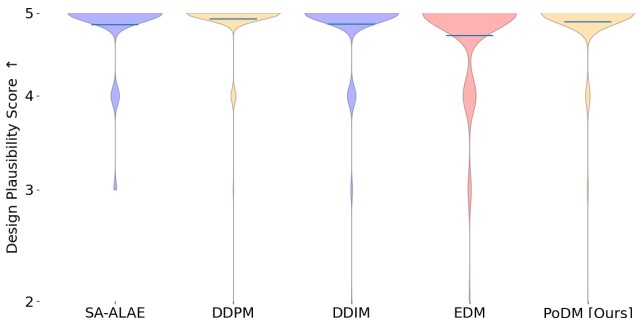

(b) On the BIKED data, we show the detailed empirical distribution of the design plausibility scores measured for each model.

Figure 5: Quantitative Evaluation of model performance.

### 4.1 Training configurations

Our work utilizes the model architecture from the DDIM (Song et al., 2020) repository for all diffusion-based models, which follows the U-Net proposed by Ho et al. (Ho et al., 2020). More precisely, the implemented model has six feature map resolutions from $256 \times 256$ to $4 \times 4$, one residual block for each

upsampling/downsampling, and an attention layer at the feature map resolution of $16 \times 16$. For sampling with DDPM and DDIM, we use the same trained model with default training settings, i.e., timesteps of $1\,000$ and linear schedule of $\beta$ with $\beta_0 = 10^{-4}, \beta_T = 0.02$. For EDM, we remove EDM's preconditions, since they did not bring much enhancement to the results according to their experiments, and implement their noise schedules for both training and sampling with default parameters, i.e., $\sigma_{\min} = 2 \times 10^{-3}, \sigma_{\max} = 80, \rho = 7, P_{\mathrm{mean}} = -1.2, P_{\mathrm{std}} = 1.2$. For our PoDM, we determine $\sigma_{\mathrm{start}} = 5.3$ and $\sigma_{\mathrm{end}} = 0.6$ by analyzing the BIKED dataset and inheriting the loss function from EDM. For stochastic sampling in both EDM and our PoDM, we allow the "churn" modification (see Equation 5) for all sampling steps, i.e., $S_{\min} = 0$, $S_{\max} = \infty$, and set the $S_{\mathrm{churn}}$ to 5. For DDIM, we use 50 as the number of sampling steps, whereas for both EDM and our PoDM, the number of sampling steps is set to 18. The set "Standardized Images" from BIKED Dataset (Regenwetter et al., 2021) contains $4\,512$ grayscale pixel-based images with original shape of $1\,536 \times 710$. We pad them with background pixels to a square form with the shape of $1\,536 \times 1\,536$. Then, we reshape these images into a resolution of $256 \times 256$ with the scale of $[-1, 1]$ in order to maintain the height-width ratio and ease the complexity in generation. From the whole dataset, we randomly select 100 images for validation, $1\,000$ images for testing, and the rest of the images for training. We run training on four NVIDIA DGX-2's Tesla V100 GPUs with a batch size of 32 and a learning rate of $5 \times 10^{-5}$. Model parameters are saved every $1\,000$ steps. If the loss converges, we keep training until $100\,000$ steps and then stop it when the denoising loss does not decrease for 20 epochs. For each model, we select the best-performing model within the saved checkpoints in the last $20\,000$ steps.

## 4.2 Evaluation procedures

In our work, we evaluate the generative models regarding sampling speed, visual quality, and design plausibility. For the sampling speed, we simply record the sampling time for generating $5\,000$ images and calculate the sampling speed in FPS (frames per second) for each model. For visual quality, we further use the $5\,000$ images generated and calculate the FID (Heusel et al., 2018) between the test images and the generated images. The measured FIDs are displayed in Figure 5a.

To quantitatively evaluate the plausibility of generated designs, we implement a human evaluation method, in which the human evaluator bypasses visual qualities (e.g., blurriness and background noise) and scores the represented design in terms of plausibility. We refer to the evaluation score as the Design Plausibility Score (DPS). In this work, the generated bicycle designs are evaluated using a five-point scoring system based on the following criteria:

- No missing fundamental part;

- No floating material or extra part;

- Every part is complete;

- Parts are connected;

- Rational positioning.

For generative model considered, we randomly select $1\,000$ samples from the $5\,000$ generated bicycle images. We shuffle all selected images and keep tracking their DPS in a manner that associates each image's score with its corresponding model. This experiment aims to prevent potential biases in the evaluation of the generated images by individual target models and to sustain a uniform evaluation standard across all images. We record the measured DPSs in Figure 5b and an average DPS for each model in Figure 5a. Besides, we calculate the Plausible Design Rate (PDR), which is the proportion of plausible designs, i.e., designs with DPS of 5, in $1\,000$ generated images.

## 4.3 Results

On the BIKED data, we first show an overview of model performance in Figure 5a, exact quantitative scores can be found in Appendix B.1: our PoDM achieves a compelling FID of **4.87**, a satisfactory DPS of **4.90**,

and a high plausible design rate of **93.5%**. DDPM (Ho et al., 2020) requires the longest sampling time of 5.26 seconds for each image but performs decently well in terms of image quality, i.e., FID of 11.77, and design plausibility, i.e., DPS of 4.93 with only 6.0% implausible outcomes. As shown in Figure 5a, EDM (Karras et al., 2022) can significantly improve the sampling speed to 2.85 FPS and even enhance the visual quality to a FID of 7.84. However, EDM performs poorly in design plausibility compared to PoDM, i.e., DPS of 4.75 and a plausible design rate of 83.4%. As seen, DDIM and EDM demonstrate a trade-off between visual quality and plausibility of generated images, whereas the DDPM leverages extremely slow sampling speed to perform decently in both aspects. We need to address that although it seems that DDPM's DPS value (= 4.93) is slightly higher than PoDM's (= 4.90), there is actually no statistical difference between them (based on a Mann–Whitney U test). For the qualitative evaluation, we plot sufficient number of synthetic bicycle designs generated by various models in Appendix B.2. Hence, we state that our PoDM method can achieve the same design plausibility, a better FID, and a much faster generation/sampling speed than DDPM.

We additionally train PoDM and EDM on Seeing3Dchair (Aubry et al., 2014) images of resolution $(128 \times 128)$ with only 100 epochs. After training, we showcase the generated designs for a visual comparison in Figure 6. Visually, PoDM generates much more plausible chairs than EDM. Measured on 1k generated images, the mean design plausibility score (DPS) and design plausibility ratio (DPR: DPS $= 5$) are: PoDM (DPS 4.65, DPR 74.5%, FID 32.15); EDM (DPS 4.13, DPR 51.5%, FID 33.48). Overall, we state that our method can *significantly improve the speed of generating structural designs while maintaining their plausibility.*

### 4.4 Alignment test

It is not surprising that most-used metrics in the community of generative modeling do not align well with human judgments and fail in evaluating the plausibility of generated designs. To demonstrate this, we visualize the correlation between human evaluation results and metrics results in Figure 7.

## 5 Controllable generation

In this section, we test the PoDM's understanding of structural design space by applying cutting-edge image editing methods, e.g., interpolation via latent space, point-based dragging and inpainting, on bicycle designs.

**Interpolation via latent space** Interpolation via latent space can be quite useful in exploring structural design space. After encoding a source data $\boldsymbol{x}(0)$ to pure noise $\boldsymbol{x}(T)$ via the forward process, the diffusion model is supposed to decode $\boldsymbol{x}(T)$ back to $\boldsymbol{x}(0)$ by utilizing a corresponding ODE (Song et al., 2021). However, in our implementation shown in Figure 8a, PoDM-motivated reconstruction has poor accuracy, which might be caused by the prioritizing strategy. We argue that it is unnecessary to conduct the forward process completely, instead, perturbed images at noise level $\sigma_{\text{start}}$ retain good reversibility. Taking images at noise level $\sigma_{\text{start}}$ as latent code allows well-performing reconstruction and interpolation, as shown in Figure 8b.

**Point-based dragging** As a novel image editing method, point-based dragging (Pan et al., 2023; Shi et al., 2023) can precisely and iteratively "drag" the handle point to a target point and the remaining parts of the image will be correspondingly updated to maintain the realism. We implement DragDiffussion (Shi et al., 2023) on BIKED images and plot the results in Figure 9a. To the best of our knowledge, our work is the first to apply point-based dragging on structural design.

**Inpainting** In an inpainting task, the generative model is tasked to generate the inpainting area to match the known part. A DDPM-based inpainting mechanism, RePaint (Lugmayr et al., 2022), has achieved the state-of-the-art performance on diffusion-based inpainting tasks by utilizing the known part as guidance at each step. We adapt RePaint to our PoDM and test it on BIKED images. The inpainting results are shown in Figure 9b.

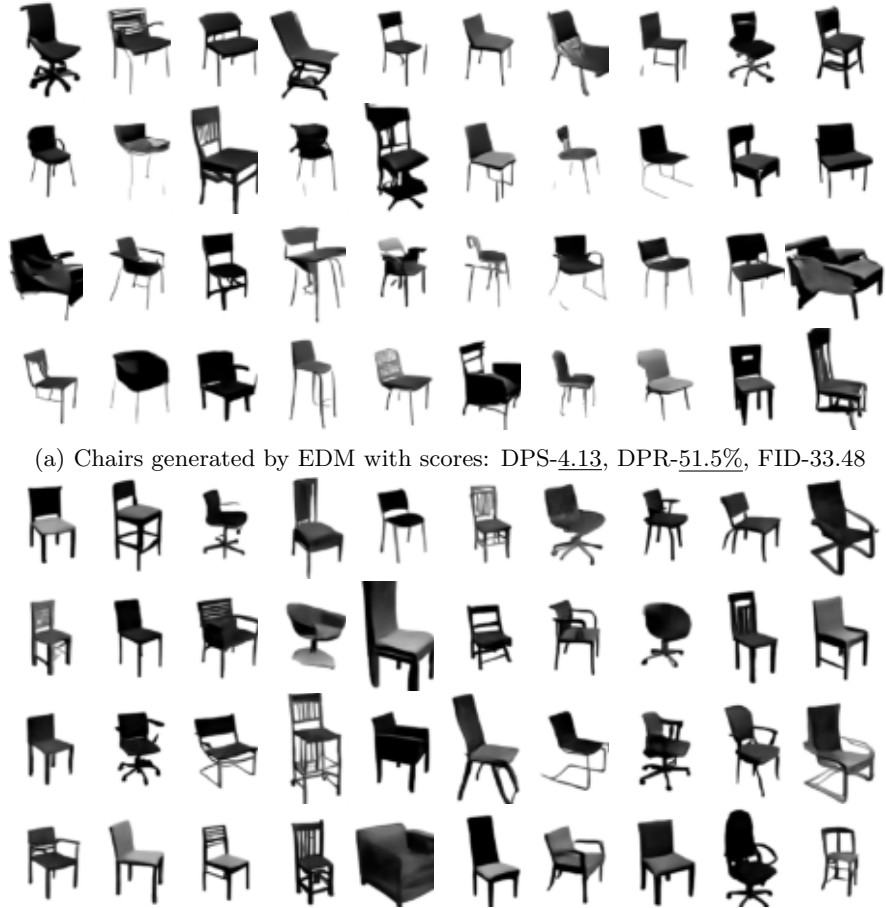

(a) Chairs generated by EDM with scores: DPS-4.13, DPR-51.5%, FID-33.48

(b) Chairs generated by our **PoDM** with scores: DPS-**4.65**, DPR-**74.5%**, FID-32.15

Figure 6: Generated chair designs with limited training epochs (100 epochs). Apparently, the images generated by EDM often have no noise, but a high structural error rate, while our PoDM can generate high-quality chair designs, and most of the chairs are structurally accurate.

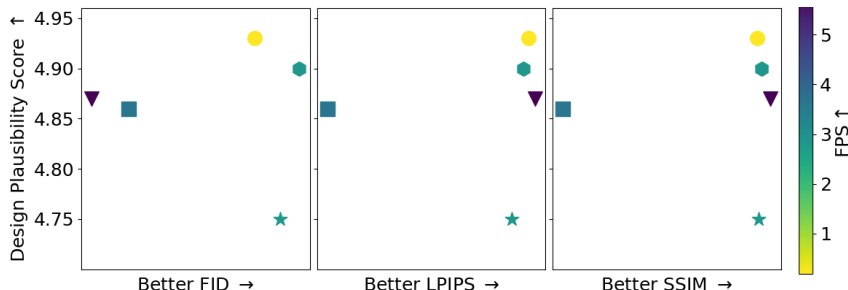

Figure 7: Alignment test. Here we test the alignment between human evaluation results (Design Plausibility Score) and various metric results. The plot shows that non of them have a strong correlation to the human evaluation.

# 6 Conclusion

We observe that the performance of the diffusion-based generative models exhibits a trade-off among visual quality, the plausibility of generated images, and sampling time. We assume that there is a range of noise lev-

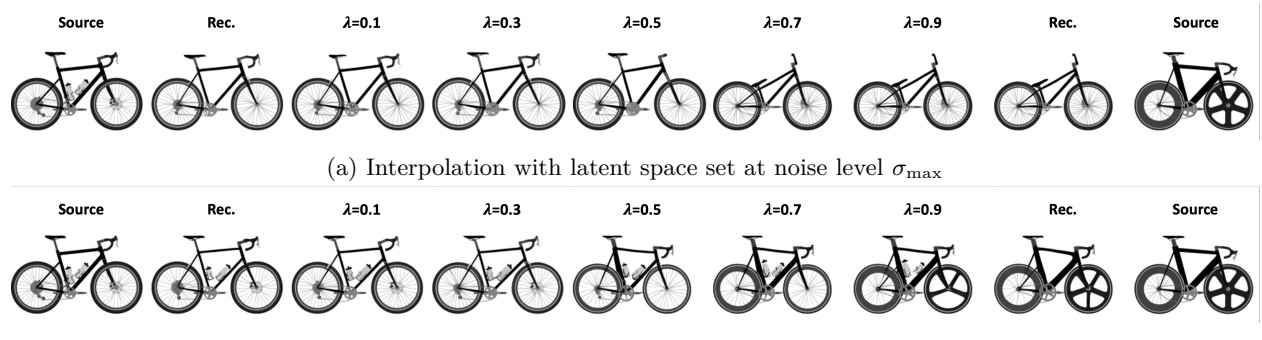

(a) Interpolation with latent space set at noise level $\sigma_{\max}$

(b) Interpolation with latent space set at noise level $\sigma_{\text{start}}$

Figure 8: PoDM-driven structural interpolation via latent space set at various perturbation steps. In (a), the reconstruction has a poor accuracy, and interpolation fails to produce intermediate structures. In (b), the interpolation displays a smooth transformation between two source structures.

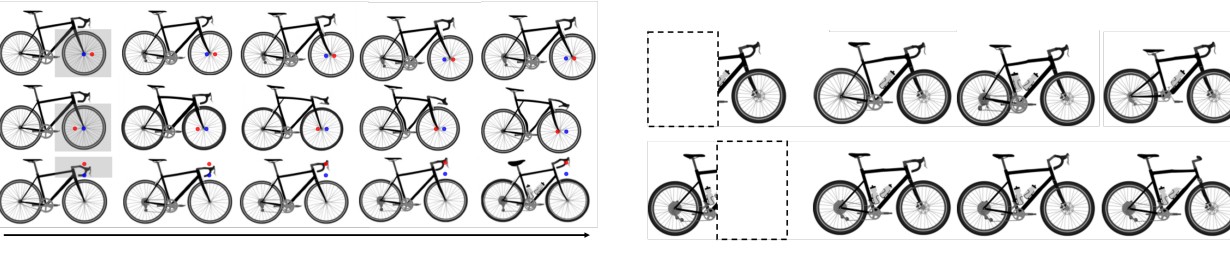

(a) PoDM-driven structure editing with DragDiffusion (Shi et al., 2023)

(b) PoDM-driven inpainting with RePaint (Lugmayr et al., 2022)

Figure 9: PoDM-driven structural interpolation via latent space set at various perturbation steps. In (a), from left to right, the handle point is iteratively dragged from the initial handle point (*blue*) towards the selected target point (*red*). In (b), the part enclosed by a *dashed line* is redesigned with RePaint.

els, that is responsible for the plausibility of the outcome, especially in generating structures. Following this observation, we propose a plausible-oriented diffusion model (PoDM) that leverages a novel noise schedule to prioritize this range of noise levels in both training and sampling procedures. We observe that the well-known EDM has a poor performance in generating plausible structures. Our PoDM method significantly improves the plausibility of generated images over EDM and also achieves a satisfactory plausibility score comparable to DDPM but with a much-reduced generation time. Additionally, we demonstrate with convincing results that the improvement in the plausibility thanks to the prioritization of the determined noise range. Further implementations of PoDM-driven image editing tools showcase PoDM's ability to semantically manipulate complex structural designs, paving the way for future work in the field of generative design.

Our work is inspired by, but not limited to, structural design generation. We believe that our observations and determinations of the phases in the diffusion process are equally applicable to images from natural scenes and, therefore, beneficial for all diffusion-based synthesis tasks. In addition, we hope that our work will inspire more research on the tool for automatically evaluating the plausibility of generated images and the relevance between noise level and generated features.

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

# A    Evolution of pixel-value distribution in perturbation experiments

## A.1    On more design data sets

In Figure 10 and Figure 11, we additionally plot the pixel-value distribution during the perturbation experiment on datasets, e.g., the Seeing3DChairs (Aubry et al., 2014) and Shoes (Yu et al., 2016).

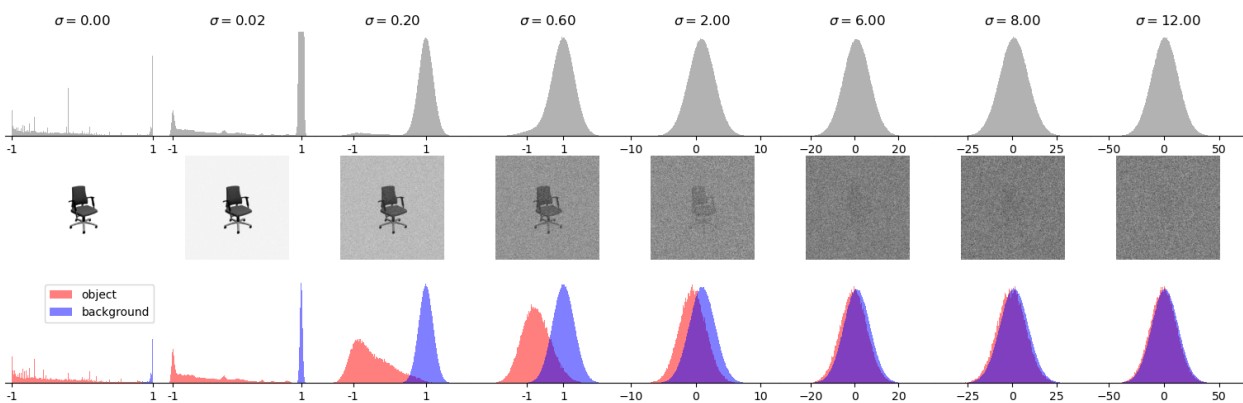

Figure 10: Evolution of the pixel-value distribution in perturbation experiments for Chair designs. In the top row, all pixel values are plotted as histograms in gray, while in the next row, the pixels are divided into background pixels and target pixels and then plotted as histograms.

## A.2    On color images

With color images in the FFHQ data set, as we do with BIKED images in Section 3.2, we track the pixel-value distribution for each color channel with object (human face) and the background separated in two distributions. In Figure 12, we plot the evolution of the pixel-value distribution in perturbation experiments. It can be clearly seen that the evolutionary process of FFHQ images is the same as the evolutionary process of the BIKED images, where the three phases can be observed.

# B    Additional evaluation results

## B.1    Quantitative evaluation

In Table 2, we list all the measured metric results of each model in terms of sampling speed, visual quality (measured with FID) and plausibility (manuelly measured).

## B.2    Qualitative evaluation

In Figure 13, Figure 14, Figure 15, Figure 16, and Figure 17, we plot a certain number of randomly generated bicycle designs for each model trained on the BIKED dataset, respectively. Here, we provide the results for the qualitative evaluation.

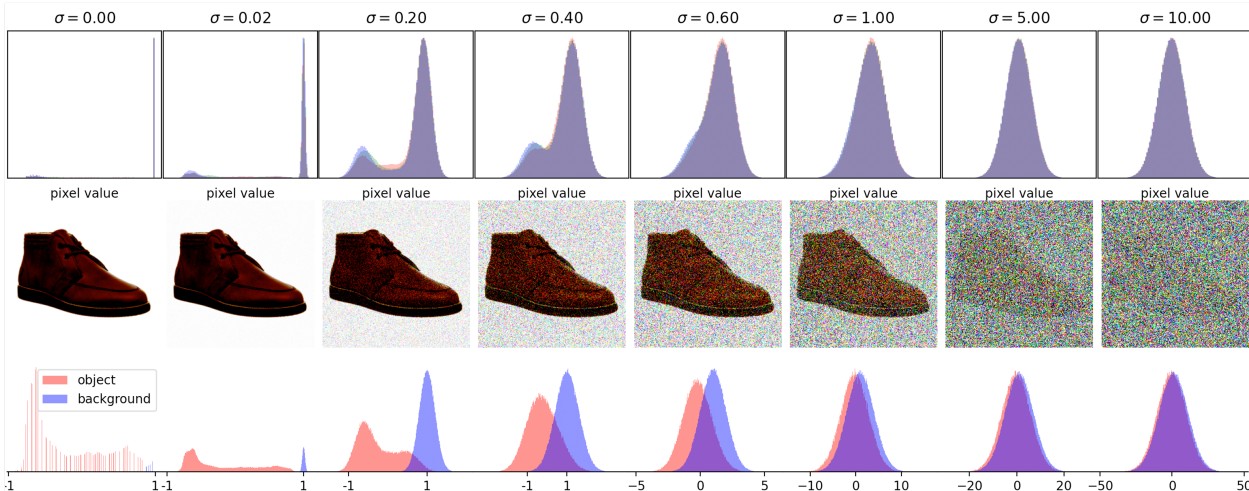

Figure 11: Evolution of the pixel-value distribution in perturbation experiments for Shoes designs. In the top row, we plot the pixel values of the three R-G-B channels as the corresponding colors; while in the bottom row, we disregard the color channels and divide them into pixel values (background) and pixel values (object). This plot shows that design images with color channels still follow our observation.

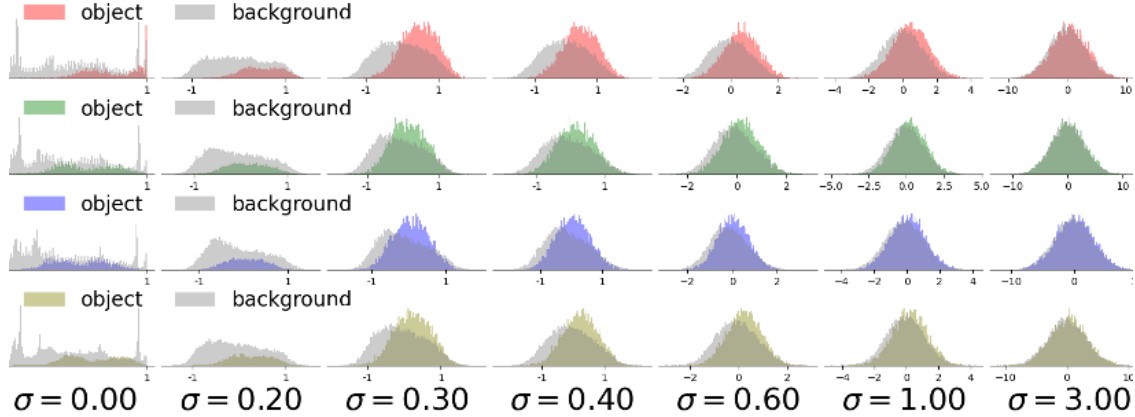

Figure 12: Evolution of the pixel-value distribution in perturbation experiments for FFHQ (Karras et al., 2019) (resolution: $64 \times 64$). Here, we manually separate the face from the background pixels. Rows from up to down: the R, G, B color channels and the grayscale.

## B.3 Parameter determining with resolution varying

EDM is proposed to elucidate the design space of diffusion models and the proposed model leverages the clean framework to achieve the state-of-the-art performance. But the EDM paper and recent work have not yet proposed a clean guidance for selecting the parameters in EDM noise scheduling. Instead, they just suggest default values $[\sigma_{\min} = 0.002, \sigma_{\max} = 80, \rho = 7, P_{\text{mean}} = -1.2, P_{\text{start}} = 1.2]$. We agree that noise scheduling should be adjusted for dataset with higher resolution, the same as other perspectives (the size of the foreground object, the difference between object and background, etc.), the question is how. Our work studies this, and proposes techniques for selecting appropriate parameters for noise scheduling to achieve better model performance. Therefor, we compare the parameters defined by our techniques with EDM default parameters.

Table 2: Quantitative comparison. Here we list the measured values of quantitative comparison. DPS: design plausibility score, PDR: plausible design rate, $r_p$: prioritization density.

| Methods | SA-ALAE | DDPM | DDIM | EDM | Our PoDM |
|---|---|---|---|---|---|
| Exponent $\rho$ | – | 1 | 1 | 7 | 7 |
| $r_p$ | – | – | – | 0.23 | **0.32** |
| FPS↑ | **5.84** | 0.19 | 3.63 | 2.85 | 2.85 |
| FID↓ | 37.05 | 11.77 | 31.35 | 7.84 | **4.87** |
| DPS↑ | 4.87 | **4.93** | 4.86 | 4.75 | 4.90 |
| PDR↑ | 89.2% | **94%** | 89.5% | 83.4% | **93.5%** |

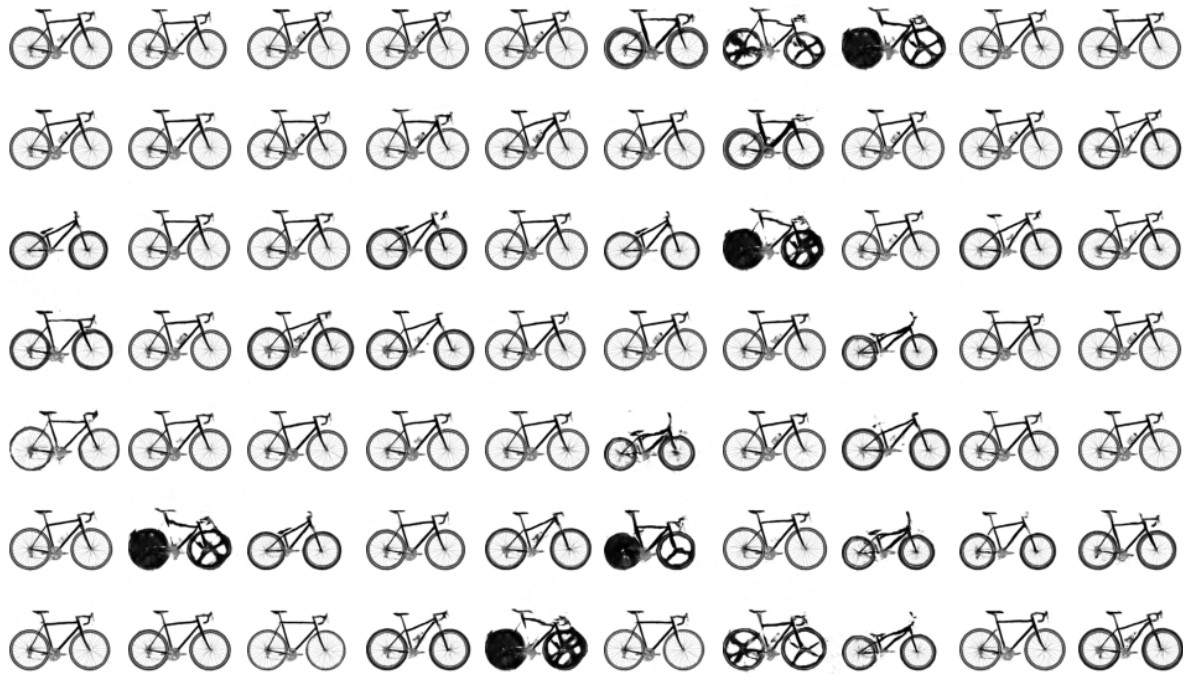

Figure 13: BIKED images randomly generated by SA-ALAE (Fan et al., 2023). SA-ALAE shows an uneven performance over various classes bikes and a great portion of generated designs are of poor quality and present implausible structures.

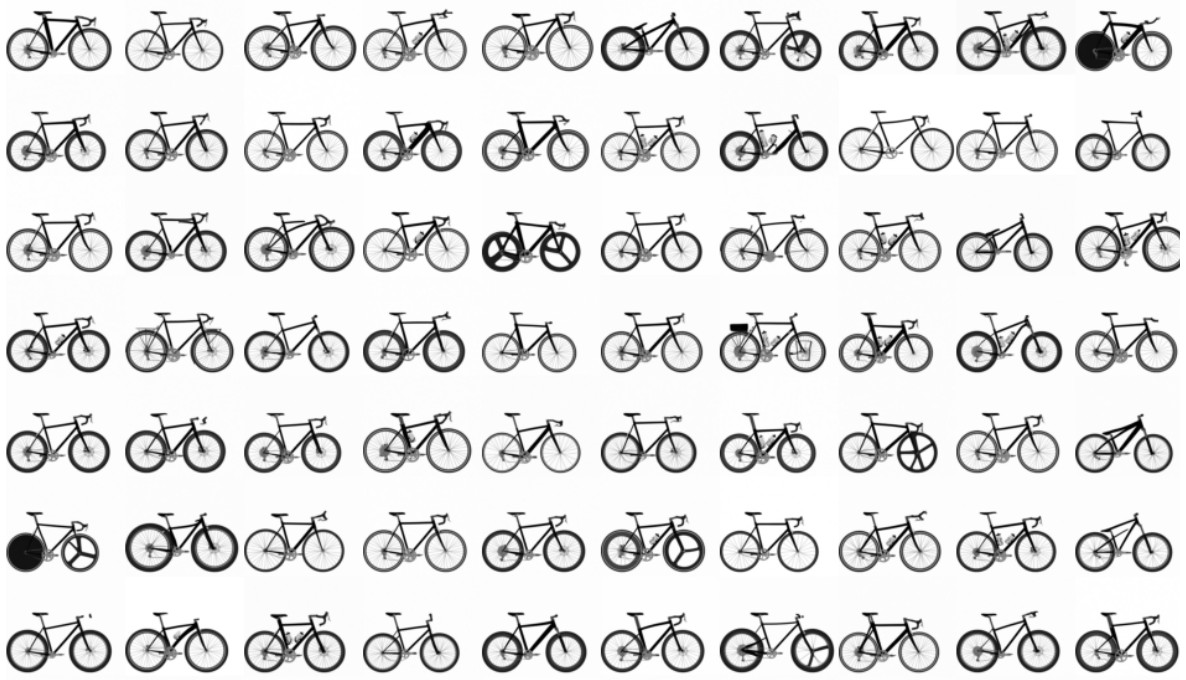

Figure 14: BIKED images randomly generated by DDPM (Ho et al., 2020). DDPM presents a strong generative power in both visual quality and structural plausibility. However, DDPM requires always a tremendous number of denoising steps (i.e., 1 000), otherwise the results look like in Figure 15.

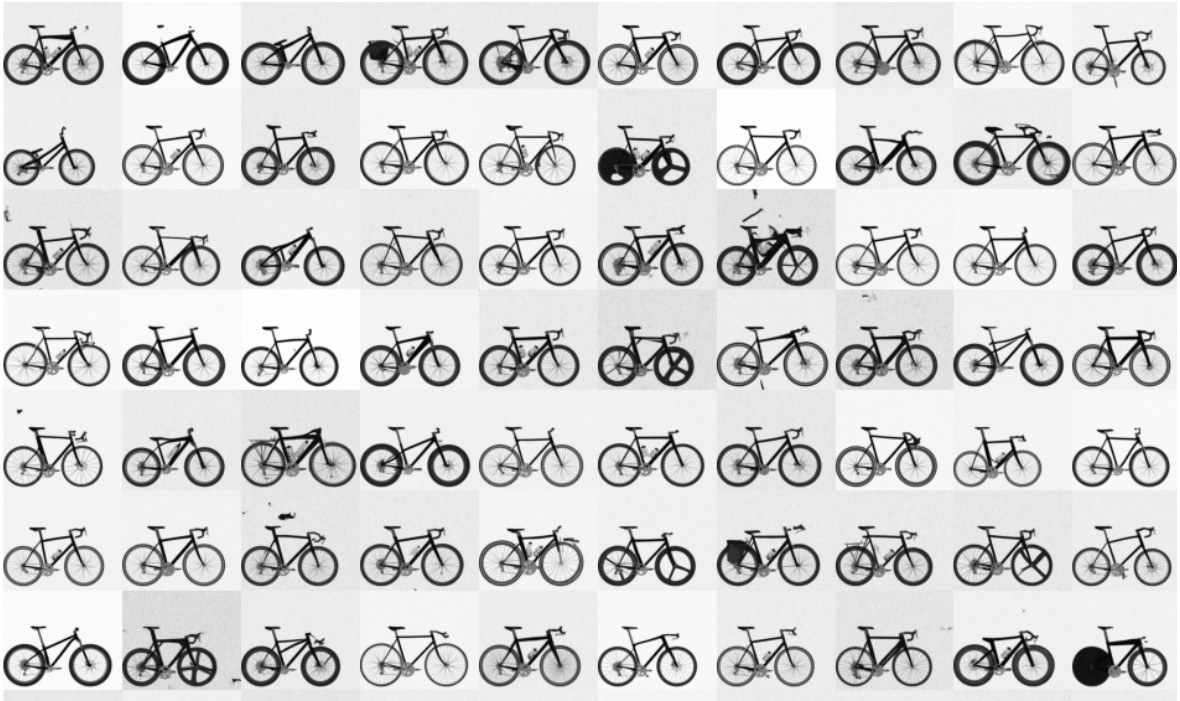

Figure 15: BIKED images randomly generated by DDIM (Song et al., 2020). DDIM leverages the same trained backbone model as DDPM, but attempts to break the Markov-chain of DDPM and to use a reduced number of denoising steps (i.e., 50). Hereby, the generated images present poor visual quality.

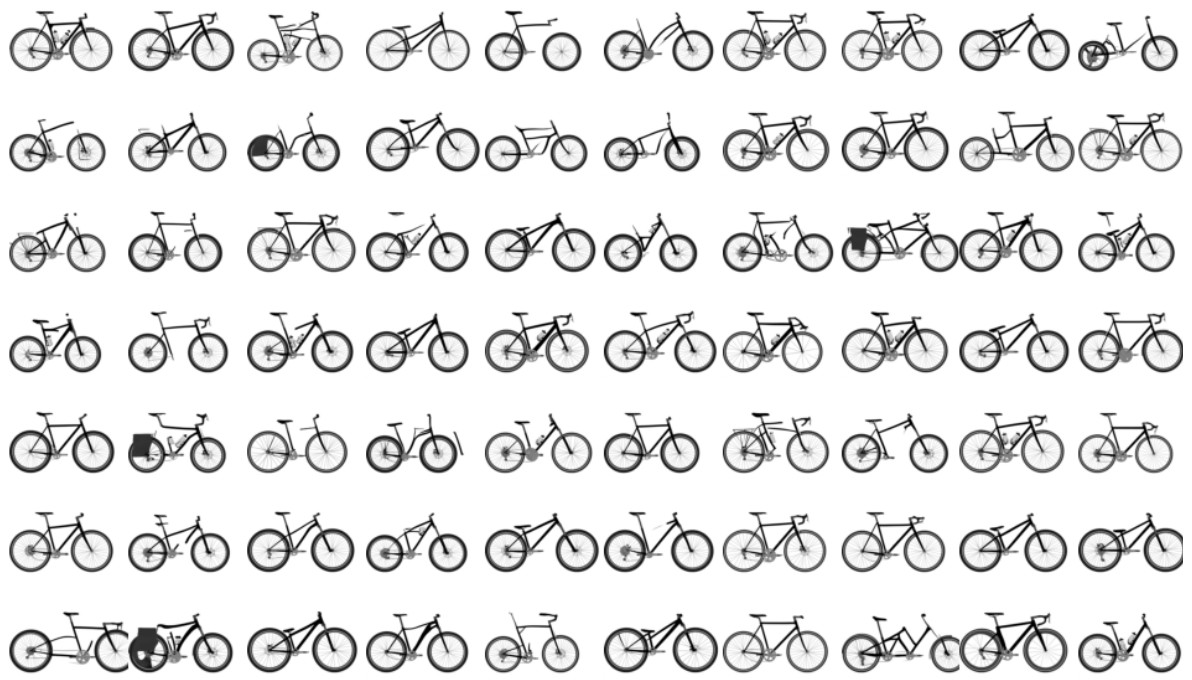

Figure 16: BIKED images randomly generated by EDM (Karras et al., 2022). EDM is theoretically based on Score-matching models (Song et al., 2021), but attempts to significantly reduce the sampling number by focusing on a pre-defined range of noise scales. Compared to DDIM, EDM has indeed deliver DDPM-like visual quality images, but we observe that it fails badly in design plausibility.

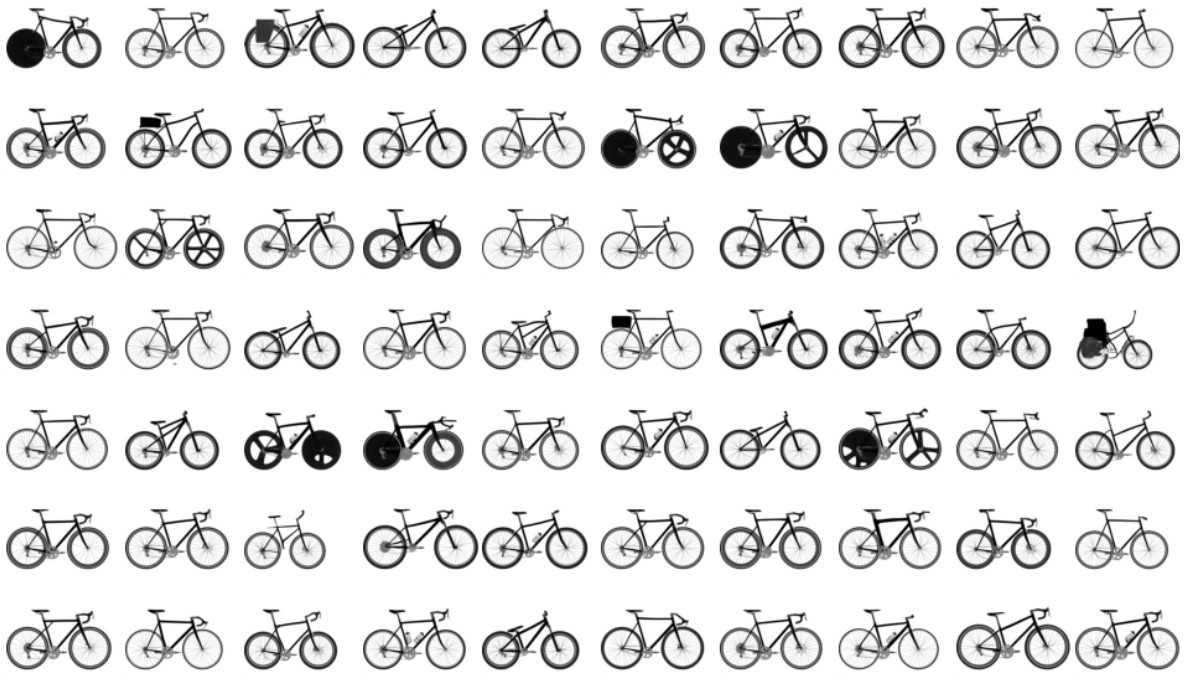

Figure 17: BIKED images randomly generated by PoDM. Based on the achievement of EDM, our work figures out a way of locating the focusing range of noise scales and hereby well-addresses the trio-trade-off among sampling time, visual quality and design plausibility.

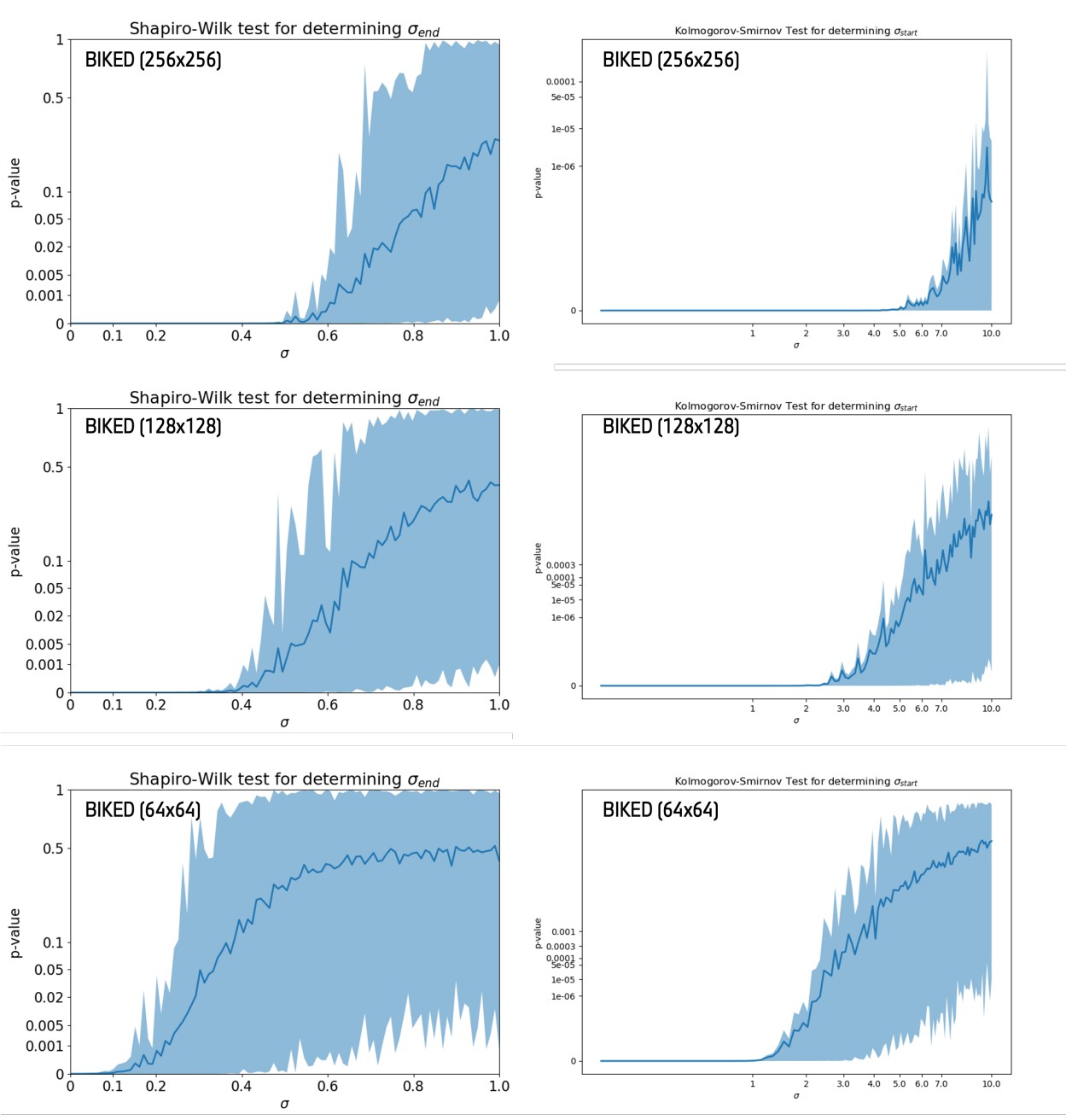

Figure 18: Parameter determining with resolution varying. With the techniques proposed by us, the determined parameters would take the resolution as well as other data characters into count, hereby delivering a reliable noise schedule.

