# OpenReview forum: "On the Efficiency of Diffusion Models in Generating Plausible Designs"
_TMLR — Rejected by TMLR_

### Review · Reviewer_rafh · 2025-04-14

**Summary Of Contributions:**

The paper introduces an approach designed to generate plausible images by prioritizing specific noise ranges during training and inference in diffusion models. The authors propose identifying a plausibility-relevant noise range through the Shapiro-Wilk test and KL-divergence and adjusting the training and sampling noise schedules accordingly. Extensive evaluations are conducted on BIKED and Seeing3DChairs datasets, demonstrating improved design plausibility, FID, and generation efficiency compared to existing methods such as DDPM, DDIM, and EDM.

**Audience:**

Yes

**Claims And Evidence:**

Yes

**Requested Changes:**

- Discuss limitations regarding object-background separation for broader datasets or more complex structures.
- Clarify how manual interventions (such as heuristic separation) influence model training and inference.

**Strengths And Weaknesses:**

Strengths:
- Clear identification and analysis of the "plausibility-relevant noise range" through statistical tests.
- Methodological contribution in adjusting noise scheduling to improve design plausibility.
- Extensive evaluation demonstrating improvements in visual quality, plausibility, and sampling efficiency.
- Comprehensive comparisons against existing methods.
- Practical demonstration of controllable generation and editing capabilities using recent image manipulation techniques (e.g., inpainting, point-based dragging).

Weaknesses:
- The proposed method relies on manual or heuristic separation of objects and backgrounds for determining plausibility-related ranges. This might somewhat limit the practicality.
- The approach is primarily demonstrated in structured designs. The paper lacks results on natural scenes with complex textures.
- The potential sensitivity of performance metrics to hyperparameter tuning (e.g., parameter $\rho$) might indicate the need for careful tuning across different datasets or applications.

---

> ### Author Response · Authors · 2025-05-02
>
> W1, W2: Indeed, separating the objects from the background is the naive way of implementing our method, but for now the most efficient way of analyzing the dataset and finding out the appropriate noise schedules. For design images with uniform background, this can be easily done; as for natural images, as we have discussed the limitation in Section 3.2 and A.2, it's not straightforward but one can use tools like segment Anything and so to finish the separating.
>
> W3: As we discussed in the paper Section 3.3, we would leave parameter $\rho$ as a tunable hyperparameter. And correct, we would also suggest to carefully tune the parameter $\rho$ in implementation, but the default value $\rho=7$ should work well on most cases.

---

### Review · Reviewer_PkeD · 2025-04-14

**Summary Of Contributions:**

This paper makes two primary contributions.

- It is a first attempt to train a diffusion model to generate plausible design images—a specific and structured type of image.

- Second, it introduces a pipeline for identifying the noise levels that best support the generation of such images.

**Audience:**

Yes

**Broader Impact Concerns:**

The paper does not include any discussion of broader impact.

**Claims And Evidence:**

No

**Requested Changes:**

- Clarify whether the shift toward higher noise levels in training is due to the nature of plausible design images or simply a consequence of using higher-resolution data (256×256). A controlled comparison or discussion would help isolate the effect.

- Add a comparison with offset noise training.

**Strengths And Weaknesses:**

### Strengths

- While noise level scheduling in diffusion models has been extensively studied, the paper demonstrates that existing schedules are not ideal for the domain of plausible design images, highlighting a domain-specific gap in prior work.

- The proposed methods for determining sigma_start and sigma_end are straightforward and easy to reproduce, making the pipeline accessible to practitioners.

### Weaknesses

- Although the abstract and introduction suggest reducing sampling time as a key motivation, the paper does not explore various well-known diffusion acceleration methods [A, B, C, D]. Incorporating these methods—or eliminating the emphasis on sampling time—could strengthen the paper’s positioning.

- The paper focuses training on higher noise levels compared to EDM, but this shift might be attributed to the higher resolution (256×256 vs. EDM’s 64×64) rather than the nature of plausible design itself. Since it is well known that higher resolutions often benefit from focusing on higher noise levels [E, F], this distinction should be clarified to better isolate the contribution.

- For image types with uniform backgrounds—as is common in the plausible design images considered—training with offset noise [G] has been shown to be highly effective. Given this, offset noise represents a strong baseline that should be included for comparison.

[A] Song et al., Consistency Models, ICML 2023.

[B] Yin et al., One-step Diffusion with Distribution Matching Distillation, CVPR 2024.

[C] Sauer et al., Adversarial Diffusion Distillation, ECCV 2024.

[D] Lu et al., Simplifying, Stabilizing and Scaling Continuous-Time Consistency Models, ICLR 2025.

[E] Hoogeboom et al., simple diffusion: End-to-end diffusion for high resolution images, ICML 2023.

[F] Esser et al., Scaling Rectified Flow Transformers for High-Resolution Image Synthesis, ICML 2024.

[G] Cross Labs., Diffusion with Offset Noise, https://www.crosslabs.org/blog/diffusion-with-offset-noise

---

> ### Author Response · Authors · 2025-05-02
>
> W1: We claim that our model is able to reduce sampling time in a way that our method needs the same sample steps as EDM (18 steps) to achieve the comparable performance in both FID and plausibility score as DDPM (1k steps). In addition, our method utilises the focus strategy from the EDM paper during training, which is also an efficient training method. In Figure 5a, you can find the plot of model performance in terms of sampling speed. Our paper's main focus is about the trade-off among sampling speed, visual quality and plausibility. Thus, this is different from reducing sampling time of diffusion models in general (the A, B, C, D references mentioned by PkeD).
>
> W2: Good point and correct. EDM is proposed to elucidate the design space of diffusion models and the proposed model leverages the clean framework to achieve the state-of-the-art performance. But the EDM paper and recent work have not yet proposed a clean guidance for changing the default parameters in EDM noise scheduling. Instead, they just suggest default values $[\sigma_{\text{min}}=0.002, \sigma_{\text{max}}=80,\rho=7,P_{\text{mean}}=-1.2, P_{\text{start}}=1.2]$, which were designed and evaluated only on images of resolution 64 (this is the limitation of paper EDM, we handle it). The question is, how should the noise schedules be when we are using it for a novel dataset with resolution that is not 64. We agree that noise scheduling should be adjusted for dataset with higher resolution, the same as other perspectives (the size of the foreground object, the difference between object and background, etc.), the question is how. Our work addresses this, and proposes techniques for selecting appropriate parameters for noise scheduling to achieve better model performance.
>
> In Figure 21 in the supplementary material as well as Figure 18 in the appendix (we just added it in revision), we showcase that our method does consider the resolution of the images. The selected values for parameters changes with the solution. And for BIKED images of resolution 64 $\times$ 64, the selected $[\sigma_{\text{end}}=0.01, \sigma_{\text{start}}=1.1]$ are quite close to the one of EDM (if we convert the default parameters of EDM into our parameter system, we have $[\sigma_{\text{end}}=0.009, \sigma_{\text{start}}=1]$).
>
> W3: A really good idea. For now, images generated by our model has already shown consistently perfect background, i.e., no floating material or being noisy, see Figure 17. Maybe because the training data for our model contains only images with uniform background, unlike the training data used for stable diffusion model. We would mention the blog in revision, and in future work have a comparison with it.

---

### Review · Reviewer_cXBx · 2025-04-20

**Summary Of Contributions:**

In the EDM framework of diffusion models, the noise schedule (indicating the noise level of each timestep $t$) is defined according to a formula involving 3 parameters, $\sigma_{min}$ (lowest noise level), $\sigma_{max}$ (highest noise level), $\rho$ (decay factor indicating how to interpolate from $\sigma_{min}$ to $\sigma_{max}$).

The reviewed manuscript proposes a simple heuristic to select these 3 parameters, based on statistical properties of an image (pixels values in foreground and background) of the dataset and grid-search.

The authors claim that their method to select these 3 parameters leads to better FID (compared to using the default value of these 3 parameters from EDM) on two datasets: BIKED and Seeing3DChairs.

**Audience:**

No

**Broader Impact Concerns:**

Same as any other work on generative modeling.

**Claims And Evidence:**

No

**Requested Changes:**

RC1/ As mentioned in W1 and W2, it is easy to imagine simple counterexamples image datasets which can break your method. Please discuss the assumptions required for the method to work and its limitations. What kinds of datasets would violate these assumptions? Please discuss the limitations of the work (including these failure cases) in detail.

RC2/ Would it be possible to experiment on more datasets, to better highlight whether the method would generalize to other datasets or whether it “overfitted” to these 2 datasets? This would also help understanding when the proposed method could fail.

RC3/ Please discuss prior works more thoroughly, and how they select the lowest and highest noise levels, including for instance (but not limited to) the 2 works mentioned in W7.

RC4/ Provide justification for the choice of hyperparameters, especially “significance level of 0.01”, “KL-divergence” threshold of “0.02”, extending the range to “99.7%”, etc. All these values seem quite arbitrary and it is unclear if other reasonable values could be used or whether these values have been optimized for the 2 datasets.

**Strengths And Weaknesses:**

Strengths:

S1/ The proposed method is relatively simple and data-driven.
The method to select $\sigma_{min}$ and $\sigma_{max}$ mostly involves looking at the distribution of pixel values for the foreground and background.
We first select $\sigma_{min}$ as the highest noise level for which the distribution of pixel values for the foreground does not look Gaussian.
We then select $\sigma_{max}$ as the lowest noise level for which the distributions of pixel values for the foreground and background are sufficiently close.
Then, the range $[ log \sigma_{min}, log \sigma_{max}]$ is widened by a factor of 3.
Once $\sigma_{min}$ and $\sigma_{max}$ are selected, $\rho$ is then selected through a grid-search.

S2/ The proposed method is original and leads to improved FID scores on 2 datasets relative to default EDM settings.


Weaknesses:

W1/ The method seems to be inherently flawed and can fail under simple counterexamples. It is easy to construct a dataset of images in which the pixel values of the foreground objects are already Gaussian-distributed or just black (eg, binarize the BIKED dataset into black and white images). This would lead to $\sigma_{min} = 0$, which is invalid.

W2/ Similarly, it could be that the distributions of pixel values in the foreground and background are already indistinguishable (potentially for natural images?), leading to $\sigma_{max} = 0$.

W3/ There is also no guarantee that $\sigma_{min}$ is lower than $\sigma_{max}$ with the proposed method.

W4/ The proposed method lacks theoretical justification. There is no formal justification or proof as to why the proposed method of selecting $\sigma_{min}$ and $\sigma_{max}$ is valid and why it should lead to better generation quality. The notion of “plausibility-relevant range of noise levels” is not clearly defined or supported.

W5/ Furthermore, the method is only evaluated on a few datasets, making it unclear if the method could generalize to other datasets, especially for datasets with different image statistics or structure.

W6/ The significance levels used for selecting $\sigma_{min}$ and $\sigma_{max}$ seems quite arbitrary, and it is not discussed how they were chosen or how they influence the result. E.g, significance level of 0.01 for Gaussianity tests or a KL-divergence threshold of 0.02.

W7/ The discussion of prior work is incomplete, for example:

Example 1: “Improved Denoising Diffusion Probabilistic Models” (Nichol, Alexander Quinn, and Prafulla Dhariwal. "Improved denoising diffusion probabilistic models." International conference on machine learning. PMLR, 2021.) selects the minimum noise levels to be “smaller than the pixel bin size 1/127.5”, which is arguably a better argument to select $\sigma_{min}$ than the proposed method, as it guarantees the distribution of clean data and noisy data (at lowest noise level) to be very close.

Example 2: “On the Importance of Noise Scheduling for Diffusion Models” (Chen, Ting. "On the importance of noise scheduling for diffusion models." arXiv preprint arXiv:2301.10972 (2023).) demonstrates that the range of noise levels decided for training should be adapted to the dimensionality of the images. The proposed method does not obey this rule.

---

> ### Author Response · Authors · 2025-05-02
>
> To correct the misunderstanding, instead of defining the $[log(\sigma_{\text{min}}), log(\sigma_{\text{max}})]$, our work designs new parameters, i.e., $\sigma_{\text{end}}$ and $\sigma_{\text{start}}$, and set their values based on the given data set.
>
> W1: It is difficult to propose a data-driven pipeline for all datasets. If facing a special dataset where the foreground objects have already Gaussian-distributed values or are just pure black, this would lead to $\sigma_{\text{end}} = \sigma_{\text{min}} = 0$. Using $\sigma_{\text{end}} = 0$ and cause infinite value of $\sigma_{\text{max}}$ in further estimation and focusing the training overly on small noise levels. To avoid this we could clip $\sigma_{\text{end}}$ to be no smaller than 0.08 (converting the parameter values of EDM to our parameters, we obtain the value $\sigma_{text{end}}=0.08$). This part has been updated in the draft, see revision.
>
> W2, W3: Cases mentioned in W2 and W3 are not common, even with natural images. We did observe the 3-phase-wise evolution of the pixel-value distribution on FFHQ images in Figure 12. If such data exists and the distributions of foreground object and the background are already undistinguishable, then there might be no foreground object at all in the image from the start point. However, as is common sense, just like commonly used open-source datasets (such as CIFAR and ImageNet), each image contains foreground objects, which are the main targets. As for the case where $\sigma_{\text{start}}$ is lower than $\sigma_{\text{end}}$, which means "pixel distribution of foreground object is indistinguishable from the one of background" happens before "pixel distribution of foreground object becomes a Gaussian distribution". Considering that noise scales needed to disturb a certain level of the feature depends on the size of the feature, the semantic difference between the features, the original distribution of feature pixels, etc. Thus, we argue that the case described in W3 is also extremely rare.
>
> RC2, W5: We conduct the training and sampling on BIKED and Chairs. For natural images We have studied the evolution of pixel-value distribution on FFHQ, Shoes, FFHQ (with face manually separated) and latent code (for testing the applicability of our method with latent diffusion model). The phase-wise evolution has been observed with all of them. Theoretically we could have test it on natural images, but the $\sigma_{\text{end}}$ and $\sigma_{\text{start}}$ we determined based on natural images, e.g., FFHQ in Figure 12, result into significantly close noise schedule as EDM, where it is not necessary to compare. But this indicates that the method we propose improves the fixed parameter settings of EDM (e.g., $\sigma_{\text{min}}, \sigma_{\text{max}}, P_{\text{mean}},P_{\text{std}}$...), which do not update with changes in various datasets, or more precisely, there is no work has been done to guide users to adjust these parameters.
>
> W4:  As for the theoretical basis, we present an insight in Figure 2, showing that focusing on a specific range can improve result quality, as the designed noise schedule aligns better with the score curve that varies with noise levels.
>
> W6, RC4: In revision, we will change the KL-divergence test to Kolmogorov-Smirnov test as it is more clear where is the value $\sigma_{\text{start}}$. We decide to adopt the advice of reviewer cXBx about the selection of "significance level of 0.01" and "KL-divergence threshold of 0.02", we have already dropped them in the revision. Instead, a better way to determine $\sigma_{\text{end}}$ and $\sigma_{\text{start}}$, is when the p-value of Shapiro-Wilk test starts diverge (here we get $\sigma_{\text{end}}$) and when the p-value of Kolmogorov-Smirnov test starts diverge (here we get $\sigma_{\text{start}}$). We have this part already modified in the paper, see the technique 1, technique 2 and Figure 4 in the revision.
>
> W7, RC3:  The two examples of prior work mentioned by reviewer cXBx are utilizing DDPM-based noise scheduling, whereas our work focuses mainly the score-matching diffusion model. The authors of EDM paper have actually elucidated the design space: the noise scheduling of those DDPM-based or so called discrete noise schedules can be converted into continuous ones (see Section B in EDM paper Appendices), so that they could compare the performance. And as result, EDM performs better. We convert DDPM noise scheduling into continuous domain results into a noise schedule (VP) similar to PoDM ($\rho = 1$), see Figure 20 in the supplementary material. In the main paper, we have already discussed that PoDM ($\rho = 1$) shows poor performance in Table 1. Simply using the minimum noise level $\beta_{min} = 0.008$ from iDDPM)or the $\sigma_{min} = 0.02$ from EDM would ignore the fact that various datasets needs different noise schedule. And the challenges brought by iDDPM has been discussed in the EDM paper as well, see Section C.3.4 of EDM paper for details.

---

> > ### Comment · Reviewer_cXBx · 2025-05-14
> >
> > The goal of the work seems to be to provide a method to select the highest and lowest levels of noise when training a diffusion model. The author claims there is currently “no guidance how to adjust them” (see supplementary, W1), and this is not the case. As I pointed out in my review (2 examples), several works discuss how to adjust the noise levels (Example 1: how to select the lowest noise level, example 2: how to adjust the noise levels to the dimensionality of the data). It is possible to find more of such works. For instance, “Common Diffusion Noise Schedules and Sample Steps are Flawed” (WACV2024) discuss that the highest noise level $\sigma_{max}$\should be as high as possible and can be infinite with the right parametrization (v-prediction). Another example, the rectified-flow noise schedule used in more recent models (eg, Stable Diffusion 3) does use noise levels ranging from 0 to infinity too. The new method in this submission proposes a new way to select $\sigma_{min}$/$\sigma_{max}$, but it does not have theoretical justification as to why the method should work. Eg, looking at the distributions of pixel values for the foreground and background, and checking when they become similar to each other does not make much sense to me as a method to select $\sigma_{max}$ ($\sigma_{start}$), compared to what one would usually do (select $\sigma_{max}$ to be infinite, or high enough so that the (multivariate) distribution of the noisy data is close enough to Gaussian).
> >
> > Misunderstanding: The answer uses two different names ($\sigma_{end}$ and $\sigma_{min}$) for the lowest noise level before and after widening the noise level range, and I used a single name in my review. I do not see any misunderstanding.
> >
> > W1: The authors claim “It is difficult to propose a data-driven pipeline for all datasets” with their method to select  $\sigma_{max}$ and $\sigma_{min}$. Given that the main goal seems to be to propose a new method to select $\sigma_{max}$ and $\sigma_{min}$, it is hard for me to understand the contribution if the proposed method only work on 2 very specific datasets.
> >
> > W2 and W3: The authors argue that the cases I describe are “not common”/” extremely rare” according to “common sense”. Yet, there is no clear argument. On the opposite, I do not understand why this would hold in general. If I take photos of objects in the wild (natural images, maybe CIFAR), it is not evident to me why the distribution of pixel values for the objects and for the backgrounds would differ (eg, in general, I don’t think the objects tend to be greener than the background, I don’t think the objects tend to be brighter than the background, etc, so why would it differ?). There is also no reason for me to believe the distribution will become Gaussian-like before becoming similar to each other, so I don’t understand why this should be “extremely rare”.
> >
> > W4: The figure 2 that the authors mention just shows the set of noise levels with the proposed method are higher than with default values of EDM. Again, there is not clear proof as to why the proposed method for selecting  $\sigma_{max}$ and $\sigma_{min}$\should work and yield improved results. It is not clear why comparing the noise levels distributions to the magnitude of the Stein score is meaningful, and, if that was really the case, why not set the distribution of the noise levels to match exactly the magnitude of the Stein score?
> >
> > W5: Again, the proposed method is only evaluated on 2 specific datasets. The discussion on pixel values in FFHD seem a bit unrelated to the proposed method as it looks at R, G, B channels instead of background/foreground. Again, the goal of the work is to provide a method to select the highest and lowest levels of noise, and there is not theoretical derivations to justify the proposed method, so extensive quantitative evaluation is necessary.
> >
> > W6: Why were these changes necessary? How do you define “diverge” and “begins to diverge” for the user to implement the proposed method?
> >
> > W7: The answer is not clear. Since these works all use different notations to refer to the same underlying “diffusion model” principle (as the authors say, they all “can be converted into” the others), it is not clear why we could not use the method to select the minimum/maximum noise level from related works with the EDM framework this work is using.
> >
> > There are typos in the revision, eg $\sigma_{textend}$.

---

> > > ### Author Response · Authors · 2025-05-16
> > >
> > > Misunderstanding exists: (1) Our work does not simply aim to select $\sigma_{\text{min}}$ and $\sigma_{\text{max}}$. The parameters setting of EDM (i.e., $\sigma_{\text{min}}, \sigma_{\text{max}}, P_{\text{mean}}, P_{\text{std}}$ and $\rho$) defines not only the sampling but also the training noise schedules, where even though a lot of recent work have studied the lowest and highest noise levels $\sigma_{\text{min}}, \sigma_{\text{max}}$, no guidance of how to adjust other parameters and how should all these parameters align with each other. Our work achieves it with $[\sigma_{\text{end}}, \sigma_{\text{start}}]$. Please check the Sec. 3.3. (2) Reviewer confused the plausibility-relevant range $[\sigma_{\text{end}}, \sigma_{\text{start}}]$ with the noise range $[\sigma_{\text{min}}, \sigma_{\text{max}}]$, please check Fig. 1 (c). We use $\sigma_{\text{end}}, \sigma_{\text{start}}$ to denote the noise range, which is assumed to be relevant to the structure modeling, from them we obtain not only $[\sigma_{\text{min}}, \sigma_{\text{max}}]$ for sampling scheduling but also $\mu$ and $\zeta$ (see Eq.8 and Eq.9) for training scheduling.
> > >
> > > W1: We studied sufficient number of datasets, BIKED, FFHQ, Latent data, Chairs, shoes, FFHQ with face manually separated.
> > >
> > > W2-W3: Our work aims to improve the plausibility of generated images, mainly the plausibility of foreground object. In Fig. 12, when separating human faces from the background, we can easily observe the huge difference between pixel-value distribution of human face and the one of background. From a sufficient amount of data, the generative model is tasked to capture the pixel-value distribution of certain object, so that it could generate a novel sample. This explains, why in many generated human face picture, human faces tend to be realistic while background are full of artifacts. For same reason, if there are 100 CIFAR images of tigers, the distribution of tiger pixels will differ from the one of background pixels. Thus, we could use our method to get the range of $[\sigma_{\text{end}}, \sigma_{\text{start}}]$ for defining the training and sampling schedules. To this end, we do not see the necessity of discussing image plausibility where no big difference between foreground objects and background ones, like a picture of sky.
> > >
> > > W4: Score-matching models uses training to simulate the score function, so that the denoising can guide the image to high-density scores. Misalignment of the focus of training and sampling with the score function causes inefficiency. However, before training score function the score function is unknown. EDM defines their training schedule based on the results from previous models, whereas our method can determine efficient schedules by analyzing the data and we observe that the resulted schedules align well with the underlying score function.
> > >
> > > W5: For FFHQ, please check Fig. 12, where we did separation of foreground and background.
> > >
> > > W6: See Fig.18, the idea of determining the $\sigma_{\text{end}}$ when the curve ``begins to diverge'' is quite trivial to implement. Since this is a technique for defining hyperparameters, we do not believe it is necessary to determine exact values.
> > >
> > > W7: The background knowledge about variance preserving, variance exploding and their corelation has been well explained in the EDM paper. Previous works have discussed about the $[\sigma_{\text{min}}, \sigma_{\text{max}}]$, but not the training part of EDM, where $[\sigma_{\text{min}}, \sigma_{\text{max}}]$ do not play a role.

---

### Author Response · Authors · 2025-05-02
**official comment to all reviewers**

Thanks for the detailed feedback from all reviewers, we will correspondingly reply all your concerns and hopefully the comments and revision could answer your concerns. We create several figures for the rebuttal and all comments to the reviewers can be found in the supplementary material. Also, we have accordingly updated several parts of our paper draft, see revision.

---

### Decision · Action_Editor_5cHR · 2025-07-31

**Recommendation:** Reject

**Audience:**

Yes

**Audience Explanation:**

Optimization of the diffusion models is a very relevant topic in contemporary machine learning.

**Claims And Evidence:**

No

**Claims Explanation:**

All the reviewers agree that the claims made in this submission is not supported by evidence.

The subject of the paper is improved noise scheduling in diffusion models. In the paper, it is proposed to use a statistical test to set the hyperparameters of the noise schedule. Empirical benchmarks show improvements stemming from the proposed methods. However, the reviewers are not convinced by the theoretical arguments of the authors nor the connection between the proposed methods and the results.